# Optimized Multi-Spectral Filter Arrays for Spectral Reconstruction

**DOI:** 10.3390/s19132905

**Published:** 2019-06-30

**Authors:** Renjie Wu, Yuqi Li, Xijiong Xie, Zhijie Lin

**Affiliations:** 1College of Information Science and Engineering, Ningbo University, Ningbo 315000, China; 2School of Information and Electronic Engineering, Zhejiang University of Science and Technology, Hangzhou 310000, China

**Keywords:** filter array design, channel selection, spectral reconstruction

## Abstract

Multispectral filter array (MSFA)-based imaging is a compact, practical technique for snapshot spectral image capturing and reconstruction. The imaging and reconstruction quality is highly influenced by the spectral sensitivities and spatial arrangement of channels on MSFAs, and the used reconstruction method. In order to design a MSFA with high imaging capacity, we propose a sparse representation based approach to optimize spectral sensitivities and spatial arrangement of MSFAs. The proposed approach first overall models the various errors associated with spectral reconstruction, and then uses a global heuristic searching method to optimize MSFAs via minimizing the estimated error of MSFAs. Our MSFA optimization method can select filters from off-the-shelf candidate filter sets while assigning the selected filters to the designed MSFA. Experimental results on three datasets show that the proposed method is more efficient, flexible, and can design MSFAs with lower spectral construction errors when compared with existing state-of-the-art methods. The MSFAs designed by our method show better performance than others even using different spectral reconstruction methods.

## 1. Introduction

Multi-spectral images are widely used in many applications such as material classification, object recognition, and color constancy. In recent decades, many snapshot spectral imaging techniques [1] have been proposed. Among these techniques, multi-spectral filter array (MSFA)—based imaging is a compact, convenient, and robust way to acquire and reconstruct spectral images. The method using a camera with mosaic of the multiple filters on the sensor has become commercially available, e.g., SILIOS technologies developed a manufacturing technique called COLOR SHADES^®^ to produce MSFAs [2].

However, most of the existing MSFAs are usually designed in an ad-hoc manner, e.g., the spectral sensitivity function of channels are designed as a series of band-pass functions with a close full width at half maximum (FWHM) and the same intervals. In fact, to design an optimal MSFA, the influence of spectral sensitivity functions of channels, channel spatial distribution, and reconstruction method are mutually correlated [3]. Previous works [4,5,6,7] optimize either (a) the spectral sensitivity functions of channels or (b) the arrangement of channels on periodic mosaic pattern. These methods treat the two issues separately thus cannot guarantee the optimal MSFA design. Besides, the MSFAs design methods do not take advantage of multispectral image statistics thus cannot assure an accurate spectral recovery. In addition, previous works optimize spectral sensitivity functions of channels based on theoretical transmittance function (e.g., radial basis function [4], Fourier basis function [8]). However, due to the limitation of manufacturing, MSFAs confirming to such theoretically perfect functions or arbitrary manually set functions are usually not available in practice [9]. Therefore, compared with theoretical optimization, selecting a few filters from a candidate commercial filter set is more practical for MSFA design.

It is worth noting that the MSFA design also depends on the applied spectral reconstruction method. Joint design of a MSFA and a spectral reconstruction algorithm is potentially effective. Here spectral reconstruction is an operation that recovering spectrum at the position of each pixel from incomplete samples output from a MSFA sensor. The most widely applied solution first obtains the full resolution image of multiple channels through multispectral demosaicing [10], and then recovers spectral information of each pixel using the obtained multiple responses [11,12,13]. Recently, some works have demonstrated the great capability of sparse coding either for color demosaicing [14,15] or for spectral recovery [16]. However, the two-step spectral reconstruction may introduce propagation error. Thus reconstruction quality can be further improved using an overall sparse representation method [17,18].

Based on these considerations, we propose a sparse representation overall model to estimating reconstruction errors with prior knowledge. We firstly introduced a sparse coding method to directly recover multispectral images from raw outputs of MSFA sensors; then we estimate and qualify the reconstruction errors of the spectral reconstruction method to evaluate the capability of MSFA sensors; finally, the optimal MSFA can be achieved by (1) selecting a subset from a large candidate set of commercial filters; (2) assigning the selected filters onto MSFA mosaic pattern that minimizes the estimated reconstruction errors using a heuristic global searching method.

## 2. Related Works

In recent decades, many efforts have been made to study training based spectral image reconstruction and filter optimization for multispectral cameras. As shown in Figure 1, these works mainly focus on the following issues:

### 2.1. Spectral Reconstruction

Spectral images can be recovered from multiple-channel raw response via learning a mapping between the spectra and the corresponding responses for a given multispectral camera. The mapping is learned from a training set consisting of the sample pairs of "spectra-responses" images. Some works pose spectral reconstruction as a traditional high dimensional interpolation problem and solve it using pseudo-inverse method [19], locally linear regression [20] or nonlinear radial-basis-function approximation [21,22].

Recently, other ways of reconstructing spectral images from raw response images (usually RGB images) have been explored, including sparse reconstruction and deep learning. Sparse reconstruction methods exploit the sparsity of spectra, recover spectra for every pixels using an learnt overcomplete dictionary [23,24,25]. Based on sparse reconstruction, some literature has proved that the reconstruction accuracy would benefit from combining the prior of local manifold structure [16,26]. To further improve reconstruction quality, the convolutional neural network based learning methods which utilize the spatial information of images have begun to appear [27,28]. Various network structures have been proposed for a contest to show the potential of deep learning in spectral construction [29]. However, none of these spectral reconstruction methods assume the raw response images are mosaicked, thus cannot be applied to our problem. A more recent literature [30] utilized random printed masks as filter arrays and reconstructed multispectral images using an end-to-end network.

### 2.2. Spectral Filters Selection

Filter selection is to select a few used filters from a candidate filter set while ensuring reconstruction accuracy. Imai et al. compared the behaviours of narrow-band and wide-band filters on spectral reflectance estimation [31]. Chi et al. minimized the condition number of spectral sensitivity of filters to make the spectral construction more robust to noises [7]. Other methods take both the prior of spectral information and imaging noises into account, e.g., Ansari et al. [32] used a Wiener filter estimation method to estimate reconstruction error of Munsell spectra; Wang et al. [33] presented a model that estimates the spectral reconstruction error of multi-spectral imaging system using the prior of spectral correlation and discussed the tradeoff between choosing narrow filters and wide filters; Shen et al. [34] also estimated the reconstruction error and applied binary differential evolution algorithm to find the optimal combination of filters. Instead of estimating reconstruction error, other methods used brute-force strategy that recovering all spectral images in training set using possible filters combination and calculating the real total reconstruction error [4,35]. However, compared with error estimation, actual calculation is much more time-consuming, especially when the number of possible filters combination is too large. In fact, filter selection is an NP-hard problem thus inefficient evaluation of solutions would make the selection unacceptable and impractical. Taking advantage of a pre-trained deep spectral reconstruction network, Fu et al. [36] used a Lasso-based channel pruning algorithm to select filters, but their method can only select filters for specific filter arrays.

### 2.3. MSFA Pattern Design

Previous works [37,38] have shown that a good channel spatial arrangement requires satisfying three properties: (i) channels should be spatially uniform to ensure robustness against image sensor defects; (ii) channel arrangement should be regular to ensure image reconstruction efficiency; (iii) channels should be neighbor-consistent (each channel has the same neighbor channel) to avoid the inconsistent optical crosstalk between adjacent pixels. The most popular design is the binary tree-based MFSA proposed by Miao et al. [38], the design strategy ensures all the above properties and is widely applied in real imaging systems [6,39]. However, the channel spatial arrangement of MSFAs are set manually thus cannot guarantee optimal design. Besides, the MSFA pattern must be designed with the filter selection and the used spectral reconstruction algorithm to achieve optimal imaging capacity.

### 2.4. Overall Design

The overall design takes all of the above factors into consideration. In the area of color imaging, Henz et al. [40] proposed an overall color filter array optimization method using deep convolutional neural networks. Similarly, Nie et al. [41] jointly optimized filters and recovered spectral images by optimizing the weights in CNN. Li et al. [42] adopted sparse representation to model the pipeline of color imaging and demosaicking, and then optimized filters arrays via minimizing mutual coherence. However, all of these continuous optimizations are not suitable for the discrete combination problem involved filter selection and channel arrangement. In contrast, other overall design methods [3,43] applied heuristic global search algorithm to find a near-optimal MSFA. Motivated by these overall design methods, we model the total errors in the pipeline of imaging and reconstruction by considering the spectral sensitivities of channels, the spatial arrangement of channels, and the statistical prior of spectral images as a whole.

## 3. Spectral Reconstruction Using Sparse Coding

Assume that a scene is captured by a camera with a c-channel MSFA sensor, a spectral image patch of the captured scene is denoted as *S*. Let the resolution of *S* be w×h×l, where *h* and *w* is the vertical and horizontal spatial resolution respectively, and *l* is the spectral resolution. The model of spectral imaging with the MSFA is as follows:(1)$$X\left(u,v\right)=\sum_{i=1}^{c}m\left(u,v,i\right)\;\sum_{j=1}^{l}f\left(i,\lambda_{j}\right)S\left(u,v,\lambda_{j}\right)$$
where X(u,v) denotes the response at spatial coordinate (u,v) on the response image patch, S(u,v,λ) denotes the spectral power distribution (SPD) function of the spectrum at spatial coordinate (u,v), f(i,λ) denotes the spectral sensitivity function of the i(i=0,1,…,c) th channel. The mask function m(u,v,i) with value 1 indicates the position (u,v) is assigned to the *i*th channel, while those with value 0 indicates the position is assigned to other channels.

The above equation can be written into matrix form:(2)x=Φs
where s denotes the whl×1 vector vectorized from the spectral image patch *S*, x denotes the wh×1 vector vectorized from the response image patch *X*, Φ denotes the wh×whl matrix simulating the imaging process from spectral images to response images. The matrix Φ combines the spectral sensitivity function f() of selected channels and the mask function m() of the channels. Once the matrix Φ is decided, the corresponding MSFA is uniquely designed. Therefore, our goal is to construct the optimal matrix Φ to ensure an accurate spectral reconstruction.

The pipeline of the spectral reconstruction method is shown in Figure 2. Taking full advantages of natural spectral image statistics, a natural spectral image patch can be sparsely represented by an overcomplete dictionary:(3)$$s=\sum_{j=1}^{q}d_{j}\alpha_{j}$$
where $D=\left(d_{1},d_{2},\ldots ,d_{q}\right)\in R^{whl\times q}$ is the dictionary containing *q* atoms, and ${\left(\alpha_{1},\alpha_{2},\ldots ,\alpha_{q}\right)}^{T}\in R^{q}$ is sparse coefficient vector that most of the coefficients in α are zeros.

The overcomplete dictionary is learned from a large training dataset. The dictionary learning can be formulated as:(4)$$\min_{\left\{D,A\right\}}||T-{D\;A||}_{2}^{2};\;s.t.\;\forall i=1,2,\ldots ,o,\;||A_{i}{\left|\right|}_{0}\leq k$$
where $T\in R^{whl\times o}$ is the training dataset consisting of *o* spectral image patches, and $A=\left(A_{1},A_{2},\ldots ,A_{o}\right)\in R^{q\times o}$ is a coefficient matrix composed of *o*
*k*-sparse vectors in its column. The optimization can be solved efficiently using k-SVD algorithm [44].

According to the theorem of compressive sensing, a spectral image patch can be recovered from a raw response image patch y by using sparse representation of the learned dictionary D and the sensing matrix Φ. Let us assume that the spectral image patches are *k*-sparse, it means the patches can be represented by no more than *k* atoms. Use Λ to denote the recovered support set consisting of the indexes of the non-zero coefficients. Note that the columns of ΦD are normalized to unit vectors.
(5)$$\min_{\hat{\alpha }}||y-\Phi D_{\Lambda }\hat{\alpha }{\left|\right|}_{2}^{2},\;s.t.\;size\left(\Lambda \right)\leq k$$
Adopting batch-OMP algorithm [45] can select the support set Λ and optimize the coefficients $\hat{\alpha }$. Then the spectral image patch can be recovered as:(6)$$\hat{s}=D_{\Lambda }\;\hat{\alpha }$$

In the optimization, due to the limitation of computation capability, the resolution of used raw image patches are usually small (e.g., 32×32). To handle large raw response images, the raw response images must be segmented into small patches with overlapping for optimization. Then, the small reconstructed spectral image patches can be fused to obtain the desired large reconstructed spectral image.

## 4. Estimation of Reconstruction Error

Most of the existing works [46,47] optimize the sensing matrix via minimizing the mutual coherence. Such mutual coherence minimization algorithms use a simple and efficient objective function to reduce correlations between the atoms in the learned dictionary. However, such algorithm may not guarantee the reconstruction error minimization in our problem, as we observed in our simulations. Unlike these works, we estimate the mean-square error (MSE) of reconstruction and treat the MSE as the objective function to be optimized. Based on the proposed sparse reconstruction method, analyzing the behavior of various noise and estimating the spectral reconstruction error are as follows.

Suppose a camera with a MSFA sensor acquires a noisy response vector $y\in R^{whl}$, given by:(7)$$y=\Phi s+e=\Phi D_{\Lambda }\alpha +e$$
where Φs is the noise-free responses, $e\in R^{whl}$ are zero-mean random variables with variances $\sigma_{e}$. To simplify the expressions, here we assume the imaging noises are signal-independent and spatial-independent. Sparse recovery for a particular response y can be treated as a two-steps process: first decide the support set for the response; and then optimize a least square problem using the selected support set. Therefore, with a given support set Λ, the coefficients $\hat{\alpha }$ can be optimized as:(8)$$\hat{\alpha }={\left(\Phi D_{\Lambda }\right)}^{+}y$$
According to Equation (Equation 5), the average spectral reconstruction MSE can be defined as the integral of the estimated error in least square optimization [3], with respect to support set. The MSE is modeled as:(9)$$\begin{array}{ccc} MSE & ≜ & \int_{\Lambda }p\left(\Lambda \right)\;E\{||D_{\Lambda }\hat{\alpha }-{s||}_{2}^{2}\}\\ & = & \int_{\Lambda }p\left(\Lambda \right)\;E\{||D_{\Lambda }\left(\hat{\alpha }-\alpha \right){\left|\right|}_{2}^{2}\}\\ & = & \int_{\Lambda }p\left(\Lambda \right)\;E\{||D_{\Lambda }{\left(\Phi D_{\Lambda }\right)}^{+}e+D_{\Lambda }\left(I-{\left(\Phi D_{\Lambda }\right)}^{+}\left(\Phi D_{\Lambda }\right)\right)\hat{\alpha }{\left|\right|}_{2}^{2}\}\\ & = & \int_{\Lambda }p\left(\Lambda \right)\;Tr\{\sigma_{e}^{2}D_{\Lambda }{\left(\Phi D_{\Lambda }\right)}^{+}{\left({\left(\Phi D_{\Lambda }\right)}^{+}\right)}^{T}D_{\Lambda }{}^{T}\\ & & +D_{\Lambda }\left(I-{\left(\Phi D_{\Lambda }\right)}^{+}\left(\Phi D_{\Lambda }\right)\right)Corr_{{\hat{\alpha }}_{\Lambda }}{\left(I-{\left(\Phi D_{\Lambda }\right)}^{+}\left(\Phi D_{\Lambda }\right)\right)}^{T}D_{\Lambda }^{T}\} \end{array}$$
where p(Λ) is the possibility of support set Λ, Tr{} denotes the trace of a square matrix defined as the sum of the elements on the main diagonal, ${\left(\right)}^{+}$ denotes pseudo-inversion, I denotes a whl×whl identity matrix, and $Corr_{{\hat{\alpha }}_{\Lambda }}$ denotes the correlation matrix of the coefficients $\hat{\alpha }$ in the support set Λ.

However, it is impractical to calculate the sum of the estimated error for all possible support sets. In order to approximately calculate the integral result, we adopt the importance sampling strategy that only takes account of the errors of the 10 most frequent support sets. The correlation matrix and the support sets are obtained as priors from dictionary learning. We will verify the accuracy of the estimated errors in Section 6.

The calculation for the estimated reconstruction error can be accomplished in milliseconds by GPU-accelerated parallel computing, while actual computing methods may take seconds [35] or hours [48] to calculate the reconstruction error of only a single 1000×1000 image. Although their methods do not use GPU-acceleration, our method is still much more acceptable and practical than previous works.

## 5. Heuristic Search for MSFA Design

Without loss of generality, to design an MSFA, we need to select *n* distinct filters from a candidate filter set F consisting of *m* filters (n≪m), and assigning these selected filters on a t×t square pattern. Here the t×t pattern is the minimum periodic array on the MSFA. Since the number of distinct filters usually decides the thickness of the MSFA, considering the manufacturing complexity and the practicality of the sensor, the number *n* must be small ($n\leq t^{2}$). We assume that the square pattern is much smaller than an image patch (t≪w and t≪h).

For any MSFA with known spectral sensitivity functions of filters and the arrangement of filters on periodic mosaic pattern, its spectral reconstruction capability can be evaluated using Equation (Equation 9). However, selecting filters from candidates and assigning the filters on sensor are compute-intensive and time-consuming. Based on such a situation, to make the problem solvable, we use a Genetic Algorithm to heuristically search for a near-optimal solution in global space. The basic definitions and manipulations of the applied Genetic Algorithm are as follows:

Individual Encoding—Each individual MSFA is represented by a set $\tau =\{idx_{1},idx_{2},\ldots ,idx_{t^{2}}\}$ consisting of $t^{2}$ indexes of selected filters in the candidate set. Each set τ contains *n* distinct numbers. We denote the sensing matrix of the MSFA τ as $\Phi_{\tau }$.

Fitness Function—The fitness function fitness() is defined using the reciprocal of the estimated reconstruction error shown in Equation (Equation 9). We denote the fitness function of the individual τ as:(10)$$fitness\left(\tau \right)=\frac{1}{MSE_{\Phi_{\tau }}}$$
where $MSE_{\Phi_{\tau }}$ is the estimated MSE error with the sensing matrix $\Phi_{\tau }$ of the individual τ. The lower the estimated reconstruction error level, the higher the fitness of the designed MSFA.

Select Operation—Selecting the $p_{s}\%$ individuals from current generation to the next generation using Roulette Wheel selection strategy. The greater the fitness function value, the higher the probability of being selected to survive.

Mutation Operation—In order to avoid trapping into the local optimal solution, as shown in Figure 3, we define three mutation operations of an individual, each mutation operation has its own execution possibility $p_{m}\%$:

(a) randomly select an unselected index value to replace a selected index value in the individual, but the unselected filter must be one of the neighbors of the replaced filter, where the distance between two filters is defined using the cosine distance between the spectral sensitivities of the filters;

(b) randomly exchange two index values in the individual to slightly adjust the distribution of filters in the individual;

(c) randomly replace the value at a position with the value at another position in the individual to slightly adjust the proportion of filters in the individual.

Crossover Operation—To enhance the diversity of populations, as shown in Figure 4, the crossover operation of two individuals is defined as exchanging the different index values stored in two individuals $\tau_{1}$ and $\tau_{2}$. The exchanging pairs are selected randomly with a possibility $p_{c}\%$. This operation actually swaps the filter combinations and filter distributions of the two individuals.

## 6. Results and Discussion

To verify our estimated error model, we randomly generated 50 different MSFAs by selecting filters from Roscolux filter set (It is available online along with their specifications as in http://www.rosco.com/filters.) and simulated to calculate the spectral reconstruction mean square error of each MSFA on CAVE [6] multispectral image dataset. Figure 5a shows a strong linear relationship between the objective function and the estimated spectral reconstruction error, confirming the high accuracy of our objective function as a predictor for spectral reconstruction error.

We evaluated the performance of the proposed heuristics search by comparing it with a random search which randomly generates MSFAs and calculating their estimated MSE for spectral reconstruction. The convergence rates of both processes are presented in Figure 5b. Since the plot of our optimization strategy achieves a low value and almost flattens out after 1000 s, thus only the optimized values of both strategies in the first 4000 s are shown. The higher convergence rate proved that the proposed search method can effectively accelerate the search.

In order to prove the efficiency and effectiveness of our MSFA design method, we evaluated our optimized MSFA by comparing the spectral construction error of our MSFA with the errors of three MSFAs designed by three state-of-art methods (Yanagi’s [43], Fu’s [36], and Arad’s [35]) in simulations on the three multispectral image datasets(CAVE [6], Harvard’s [49], and ICVL’s [24]). It is worth noting that for fair comparisons, we did not compare our designed MSFAs with SILIOS and other commercial MSFAs since their candidate filter set is unknown. To the best of our knowledge, Yanagi’s method is the only overall MSFA design method based on training data, and other two methods select filters for Bayer patterns without pattern design. In addition, the spectral sensitivity functions and the number of filters used by Yangi’s and other two MSFAs are quite different (e.g., nine narrow-band-pass filters used by Yanagi’s, three wide-band filters in other twos). Therefore, we designed MSFAs using corresponding candidate filter sets and compared our optimized MSFAs with their solutions respectively.

The training set we used is 100,000 spectral image patches selected from each dataset using volume maximization strategy [13]. Applying K-svd algorithm, we learned a multispectral patch dictionary consisting of q=3000 atoms for sparse reconstruction, where each atom is a whl×1 vector. The spectral resolution *l* is set to 31 in the range of visible wavelengths from 400 to 700 nm, and we set w=h=8 for 2×2 MSFA pattern in the comparisons with Fu’s and Arad’s MSFAs while setting w=h=9 for 3×3 MSFA pattern in the comparison with Yangi’s MSFA. In the genetic algorithm, we used the population size 5000 and the algorithm usually converged after 30th generations. The possibility of selection, crossover, and mutation operations are empirically set to $p_{s}\%=0.7$, $p_{c}\%=0.5$, and $p_{m}\%=0.15$. Our MSFA is optimized under moderate noise level with the standard deviation $\sigma_{e}=0.03$.

All spectral images were reconstructed using batch-omp algorithm [45] with sparsity = 10. We reconstructed each image under the noiseless case and noisy case (in presence of added noise SNR≈30 db). We ran the experiments on a 3.6 GHz Intel Core i7 workstation with an NVIDIA TITAN Xp GPU and 64 G main memory.

First, we compared our method with Yanagi’s method. We used both of methods to choose nine filters from a candidate set consisting of 31 uniformly distributed narrow band-pass functions from 400 nm to 700 nm, with 10 nm width; and assigned the nine filters onto 3×3 filter patterns. The two designed MSFAs are shown in Figure 6. Taking only five minutes, our method can design an MSFA which is close to the designed MSFA of Yanagi’s method, while their method took 20 h. We also showed the reconstruction error and the PSNR of the reconstructed images of the two designed MSFAs in Table 1. Our designed MSFA shows superior results to Yanagi’s results irrespective of the dataset used. It is worth noting that the patterns of the two MSFAs match, thus we simulate to calculate reconstruction errors using other MSFAs with the same spectral sensitivities but a different spatial arrangement. We found no better spatial arrangement for the used channels of both methods. It is evident that both of the methods can design the optimal pattern.

Then we compared our method with Fu’s and Arad’s method. We selected three channels from a candidate set consisting of RGB channels of 28 off-the-shelf digital cameras [50], and arranged the three channels on 2×2 MSFAs. Spectral sensitivities and spatial arrangement of the two designed MSFAs are shown in Figure 7. Note that although the three patterns match, our pattern is obtained by our optimization method while their patterns are manually set, thus our searching space is much larger than theirs. Besides, our method took only 10 min to achieve the results while Arad’s method consumed 10 h. To quantify the reconstruction accuracy of multispectral images, we calculated the MSE and PSNR of the reconstructed spectral images of the three designed MSFAs and presented the results in Table 2. It can be observed that our designed MSFA showed superior results to their results, especially in the noisy case.

For visualization, we also show three example images from CAVE’s, Harvard’s dataset and ICVL’s dataset and the normalized absolute reconstruction RMSE of each pixel in Figure 8a. A spectral image was reconstructed from all of its overlap patches by filling in the patches from left to right, top to bottom and fusing adjacent patches. It is worth noting that the absolute reconstruction errors of our MSFA are much more uniform in the spatial domain. The comparison of three examples of reconstructed spectra by different MSFAs is presented in Figure 8b. Although our results are not perfect, they are the closest to the ground truth (please zoom in for details), The imperfect results are due to the lack of freedom of filter selection since we chose filters from the filters of 28 very similar commercial cameras.

We also evaluated the MSFAs by using deep-learning-based spectral reconstruction method, such as the HSCNN method [27], which won the NTIRE spectral reconstruction contest [29] recently. Our designed MSFA still outperformother MSFAs although the reconstruction method has been changed. The results are shown in Table 3.

The two experiments above show that our MSFA design method is more efficient and effective than the state-of-the-art methods, irrespective of the candidate filters and dataset used, thus our method can be applied to design MSFAs using much larger candidate filter sets.

## 7. Conclusions

In this paper, we introduced an MSFA design method based on sparse representation. The proposed method takes full advantage of spectral image statistics and optimizes filter selection and spatial arrangement. We presented an analysis of spectral reconstruction error to describe the imaging and reconstruction capability of different MSFA. Through globally searching the MSFA which has the minimum reconstruction error, the near-optimal MSFA can be achieved. Experiments demonstrate that our method is robust to different image datasets and filter sets; our optimized MSFA outperforms that of other methods and can provide valuable references or guidance for MSFA designers.

## Figures and Tables

**Figure 1 sensors-19-02905-f001:**
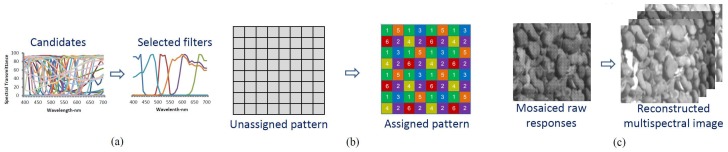
Three mutually dependent issues in MSFA design. (**a**) Selecting the used filters from commercial candidate filter set; (**b**) Assigning the selected channels on Multispectral filter array (MSFA) pattern; (**c**) Spectral reconstruction from mosaiced raw response images to reconstructed multispectral images.

**Figure 2 sensors-19-02905-f002:**
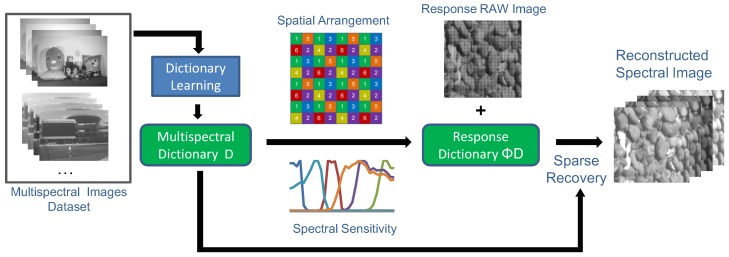
The flowchart of our spectral reconstruction.

**Figure 3 sensors-19-02905-f003:**
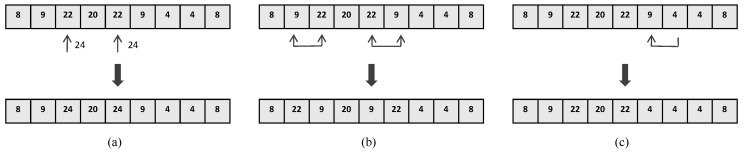
Three mutation operations. (**a**) Replace a selected filter with an unselected filter; (**b**) Exchange two selected filters; (**c**) Replace a channel with another selected channel.

**Figure 4 sensors-19-02905-f004:**
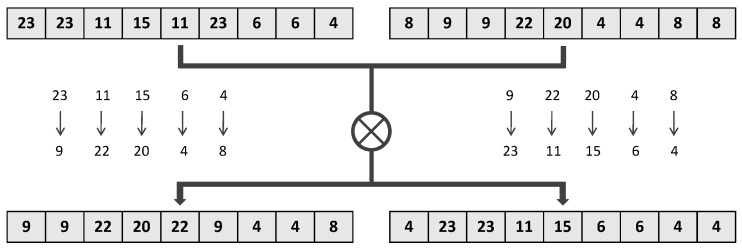
The crossover operation of two individuals.

**Figure 5 sensors-19-02905-f005:**
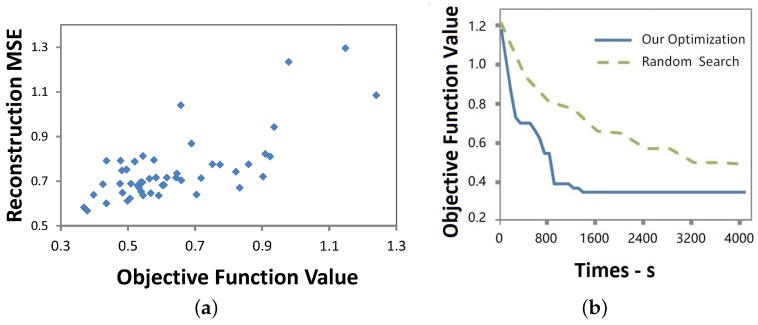
Estimated and actual errors Convergence rate (**a**) Relationship between objective function values and the actual reconstruction mean square error; (**b**) Comparison of convergence rates of the proposed search method and random search.

**Figure 6 sensors-19-02905-f006:**
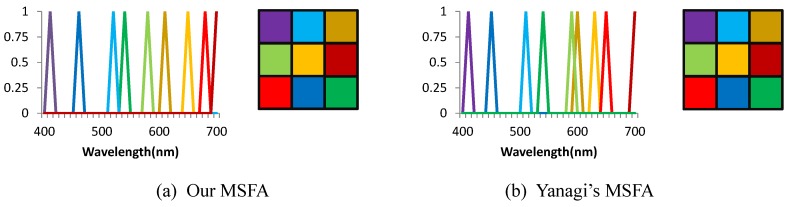
The spectral sensitivity functions and channel spatial arrangement of two MSFAs designed by Yanagi’s method and our method.

**Figure 7 sensors-19-02905-f007:**
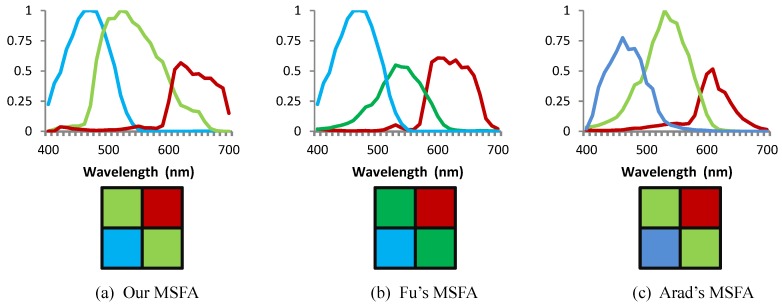
The spectral sensitivity functions and channel spatial arrangement of three MSFAs designed by Arad’s, Fu’s and our method.

**Figure 8 sensors-19-02905-f008:**
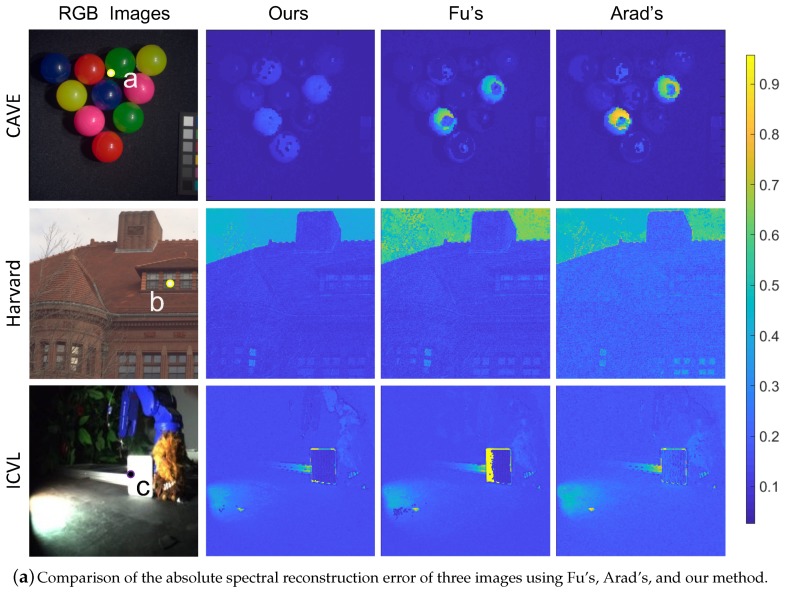
The comparison of reconstruction accuracy of the three MSFAs (see Figure 7).

**Table 1 sensors-19-02905-t001:** Quantitative evaluation of the reconstructions using Yanagi’s and our MSFA (see Figure 6) on synthetic datasets. Note that the bold font indicates the minimal error.

	MSFA	Dataset					
		CAVE		Harvard		ICVL	
		RMSE	PSNR	RMSE	PSNR	RMSE	PSNR
Noisy (SNR = ∞)	Ours	**0.172**	**21.38**	**0.165**	**22.97**	**0.215**	**19.80**
	Yanagi’s	0.203	20.67	0.190	21.18	0.268	18.42
Noisy (SNR ≈ 30 db)	Ours	**0.198**	**20.19**	**0.184**	**21.26**	**0.244**	**19.15**
	Yanagi’s	0.231	19.39	0.217	19.90	0.293	17.43

**Table 2 sensors-19-02905-t002:** Quantitative evaluation of the sparse reconstructions using Fu’s, Arad’s and our MSFA (see Figure 7) on synthetic datasets. Note that the bold font indicates the minimal error.

	MSFA	Dataset					
		CAVE		Harvard		ICVL	
		RMSE	PSNR	RMSE	PSNR	RMSE	PSNR
Noisy (SNR = ∞)	Ours	**0.258**	**20.21**	**0.212**	**21.97**	**0.071**	**28.59**
	Fu’s	0.322	19.89	0.219	21.63	0.086	28.57
	Arad	0.386	19.75	0.232	21.14	0.094	28.51
Noisy (SNR ≈ 30db)	Ours	**0.281**	**17.86**	**0.246**	**17.97**	**0.095**	**24.26**
	Fu’s	0.328	16.36	0.257	16.77	0.109	22.33
	Arad	0.440	15.32	0.288	16.03	0.156	20.98

**Table 3 sensors-19-02905-t003:** Quantitative evaluation of the deep-learning-based reconstructions using Fu’s, Arad’s and our MSFA (see Figure 7) on synthetic datasets.

	MSFA	Dataset					
		CAVE		Harvard		ICVL	
		RMSE	PSNR	RMSE	PSNR	RMSE	PSNR
Noisy (SNR = ∞)	Ours	**0.148**	**24.46**	**0.091**	**29.96**	**0.054**	**31.12**
	Nie’s	0.166	23.76	0.114	27.07	0.068	29.87
	Arad	0.161	23.54	0.119	26.49	0.069	29.89
Noisy (SNR = 30 db)	Ours	**0.163**	**23.08**	**0.097**	**27.23**	**0.059**	**30.02**
	Nie’s	0.172	22.59	0.118	25.72	0.072	27.93
	Arad	0.186	22.47	0.122	25.06	0.072	28.28

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
