# Peer review of "Optimized Multi-Spectral Filter Arrays for Spectral Reconstruction"

_sensors, 2019, doi:10.3390/s19132905_

Round 1

Reviewer 1 Report

The work is very interesting. I suggest the publication of it in the present form.

Author Response

We thank the helpful comments of the reviewer.

Reviewer 2 Report

This paper proposed MSFA imaging technique based on a sparse representation optimization of MSFAs. They also demonstrated experimental results on three datasets and they showed their method exhibits better performances compared with existing state-of-the-arts methods. It is interesting to show new method to optimize MSFA with lower spectral construction errors. I recommend this paper can be published in Sensors if some following comments are well revised.

Line 132-136, "most of the existing works~. Unlike these works, we estimate the mean-square errors~" These sentences comment on the previous other works too briefly. For these statements, please explain detailed pro and cons of authors' method and the existing works. 

Line 145-147, authors mentioned that "the proposed method takes milliseconds.Actual computing methods may take seconds[28] or hours[41]." In this comparison, there are no detailed information of previous works. For the fair comparison, authors should comment "how to estimate the process time, which calculation actual computing methods are used,..."

In fig.8, line 238-244, to be honest, it is hard to say that the proposed method (red) is the closest to the ground truth. Please present the proposed method is better than other methods in Fig. 8.

Author Response

Line 132-136, "most of the existing works~. Unlike these works, we estimate the mean-square errors~" These sentences comment on the previous other works too briefly. For these statements, please explain detailed pro and cons of authors' method and the existing works. 

Refer to Line 138-142 in the revised version

Line 145-147, authors mentioned that "the proposed method takes milliseconds.Actual computing methods may take seconds[28] or hours[41]." In this comparison, there are no detailed information of previous works. For the fair comparison, authors should comment "how to estimate the process time, which calculation actual computing methods are used,..."

Refer to Line 154-158 in the revised version

In fig.8, line 238-244, to be honest, it is hard to say that the proposed method (red) is the closest to the ground truth. Please present the proposed method is better than other methods in Fig. 8.

Explain at Line 257-260  in the revised version

Reviewer 3 Report

This article considers the problem of designing the best SFA (and CFA) sensor design. It is interesting because it includes spectral sensitivies AND spatial arrangement. it is also based on the experience of a team (in particular one of the author Yuqi Li has been very active in the field in the recent years).

As it is, the article is OK, though little difficult to read due to a few english edits. I ask for some revisions mostly because I would like a few key references to be incorporated and discussed.

It would be interesting to also discuss how the optimal filter is practically usable (in terms of real imaging cases, and how scene dynamic range is impacting the results).

Fig 5 must be improved (name of axis, values), but also it would be interesting to see how the random search converge to the same values: in optimisation, converging fast often leads only to a near-minimum (though it is not mandatory). Also the regression line does not really mean much.

Why did not you keep Monno, Wang, IMEC and SILIOS sensors in the test? though they might be non-optimal it would have been very interesting to compare to them. This is especially true for the commercial products.

Few references article miss in the article:

Compressive sensing formulation was used in related work (in particular Agarwal and Sadeghipoor), I think those should be discussed in relation:

Z. Sadeghipoor, Y. M. Lu and S. Süsstrunk, "A novel compressive sensing approach to simultaneously acquire color and near-infrared images on a single sensor," 2013 IEEE International Conference on Acoustics, Speech and Signal Processing, Vancouver, BC, 2013, pp. 1646-1650.

Marco F. Duarte, Richard G. Baraniuk, Spectral compressive sensing, Applied and Computational Harmonic Analysis, Volume 35, Issue 1, 2013, Pages 111-129,

H. K. Aggarwal and A. Majumdar, "Compressive sensing multi-spectral demosaicing from single sensor architecture," 2014 IEEE China Summit & International Conference on Signal and Information Processing (ChinaSIP), Xi'an, 2014, pp. 334-338.

Demultiplexing visible and Near-infrared information in single-sensor multispectral imaging. Sadeghipoor, Zahra; Thomas, Jean-Baptiste; Süsstrunk, Sabine, Color and Imaging Conference, 24th Color and Imaging Conference, pp. 76-81(6)

Optimal band selection and spectral reconstruction is also considered in the following works and miss in the article to track the history:

Comparison of spectrally narrow-band capture versus wide-band with a priori sample analysis for spectral reflectance estimation, Imai, Francisco H.; Rosen, Mitchell R.; Berns, Roy S. Color and Imaging Conference, 8th Color and Imaging Conference Final Program and Proceedings, pp. 234-241(8)

Imai, Spectral Estimation Using Trichromatic Digital Cameras, Proceedings of the International Symposium on Multispectral Imaging and Color Reproduction for Digital Archives, (1999), 42-49, 1999

Spectral band Selection Using a Genetic Algorithm Based Wiener Filter Estimation Method for Reconstruction of Munsell Spectral Data, K Ansari, JB Thomas, P Gouton, Electronic Imaging 2017 (18), 190-193

This recent paper may be of interest to the authors:

Hyperspectral Imaging With Random Printed Mask

Y Zhao, H Guo, Z Ma, X Cao, T Yue, X Hu - Proceedings of the IEEE CVPR, 2019

Author Response

It would be interesting to also discuss how the optimal filter is practically usable (in terms of real imaging cases, and how scene dynamic range is impacting the results).

The filter data we used is from commercial cameras, so it is able to manufacture such optimized MSFAs. We used the real spectral transmission of filters and the  ground truth of multispectral images, so scene dynamic range should not be a problem in our simulation.

Fig 5 must be improved (name of axis, values), but also it would be interesting to see how the random search converge to the same values: in optimisation, converging fast often leads only to a near-minimum (though it is not mandatory). Also the regression line does not really mean much.

Refer to Line 202-204 in the revised version

Why did not you keep Monno, Wang, IMEC and SILIOS sensors in the test? though they might be non-optimal it would have been very interesting to compare to them. This is especially true for the commercial products.

Refer to Line 209-211 in the revised version

All the necessary references are added.